# Improving our understanding of the social determinants of mental health: a data linkage study of mental health records and the 2011 UK census

Lukasz Cybulski ,[1] Natasha Chilman [1] Amelia Jewell,[2] Michael Dewey,[3] Rosanna Hildersley,[1] Craig Morgan,[3] Rachel Huck,[4] Matthew Hotopf,[1] Robert Stewart,[1] Megan Pritchard,[5] Milena Wuerth,[1] Jayati Das-Munshi [1]

¹Department of Psychological Medicine, King's College London, Institute of Psychiatry Psychology and Neuroscience, London, UK
²South London & Maudsley NHS Foundation Trust, London, UK
³Health Service and Population Research Department, Institute of Psychiatry, Psychology and Neuroscience, King's College London, London, UK
⁴Office for National Statistics, London, UK
⁵University of East Anglia Norwich Medical School, Norwich, UK

**Correspondence to**
Dr Jayati Das-Munshi;
jayati.das-munshi@kcl.ac.uk

## ABSTRACT

**Objectives** To address the lack of individual-level socioeconomic information in electronic healthcare records, we linked the 2011 census of England and Wales to patient records from a large mental healthcare provider. This paper describes the linkage process and methods for mitigating bias due to non-matching.

**Setting** South London and Maudsley NHS Foundation Trust (SLaM), a mental healthcare provider in Southeast London.

**Design** Clinical records from SLaM were supplied to the Office of National Statistics for linkage to the census through a deterministic matching algorithm. We examined clinical (International Classification of Disease-10 diagnosis, history of hospitalisation, frequency of service contact) and socio-demographic (age, gender, ethnicity, deprivation) information recorded in Clinical Record Interactive Search (CRIS) as predictors of linkage success with the 2011 census. To assess and adjust for potential biases caused by non-matching, we evaluated inverse probability weighting for mortality associations.

**Participants** Individuals of all ages in contact with SLaM up until December 2019 (N=459 374).

**Outcome measures** Likelihood of mental health records' linkage to census.

**Results** 220 864 (50.4%) records from CRIS linked to the 2011 census. Young adults (prevalence ratio (PR) 0.80, 95% CI 0.80 to 0.81), individuals living in more deprived areas (PR 0.78, 95% CI 0.78 to 0.79) and minority ethnic groups (eg, Black African, PR 0.67, 0.66 to 0.68) were less likely to match to census. After implementing inverse probability weighting, we observed little change in the strength of association between clinical/demographic characteristics and mortality (eg, presence of any psychiatric disorder: unweighted PR 2.66, 95% CI 2.52 to 2.80; weighted PR 2.70, 95% CI 2.56 to 2.84).

**Conclusions** Lower response rates to the 2011 census among people with psychiatric disorders may have contributed to lower match rates, a potential concern as the census informs service planning and allocation of resources. Due to its size and unique characteristics, the linked data set will enable novel investigations into the relationship between socioeconomic factors and psychiatric disorders.

## STRENGTHS AND LIMITATIONS OF THIS STUDY

⇒ This is the first time mental healthcare electronic records have been linked to the Office of National Statistics census at the individual-level in England. Due to its scale, ethnic diversity and demographic characteristics and abundance of detailed information on a variety of socioeconomic and demographic indicators acquired through the linkage to census records, this data set will enable novel investigations into the causes, trajectories and outcomes of psychiatric disorders.

⇒ A significant strength of the study is that we could assess and adjust for potential biases caused by non-matching related to age, gender and deprivation.

⇒ While we observed differences between individuals that matched to census, and those that did not, our weighted analyses were able to show that these differences did not substantially alter associations with mortality outcomes.

⇒ Due to the nature of the deterministic linkage algorithm, we could not determine the causes of non-linkage.

## INTRODUCTION

The growing size and depth of routinely collected administrative data available for research is transforming the study of mental disorders. Traditional epidemiological methods, such as prospective cohort or case–control studies, can present considerable methodological, logistical and financial challenges due to a high degree of attrition,[1] the inherent difficulties in selecting controls,[2] and the costs associated with data collection. Electronic health records (EHRs) and other administrative data from public services are therefore increasingly being used in epidemiological investigations because they partially address the issue of data loss by collecting information from all individuals who interact

with services.[3] They also provide a convenient mechanism for sampling controls and eliminate the need for data collection. However, despite their strengths, EHRs typically contain limited information on socioeconomic characteristics at the individual level. Data on occupational classification, long-term unemployment, ethnicity, housing tenure, education, migration and other relevant socioeconomic measures are often either missing, inaccurate or collected infrequently, hindering efforts to better understand relationships between mental health and socioeconomic and socio-demographic factors. In prior EHR research, the influence of social determinants has largely been assessed through area-level measures of deprivation, which may not accurately correspond to an individual's socioeconomic circumstances, potentially biasing observed associations and obfuscating inferences that can be made.

To address these issues, we linked clinical records from the South London and Maudsley (SLaM) Mental Health Trust accessed through its Clinical Record Interactive Search (CRIS) platform, to administrative records from the 2011 population census for England and Wales. The modern census of England and Wales, organised and conducted by the Office for National Statistics (ONS),[4] is a rich source of information on a multitude of socioeconomic indicators such as ethnicity, religion, education, employment, housing, migration and citizenship and also includes self-rated measures of health and functioning. Because of the size and the considerable ethnic diversity of the mental health services' catchment area from which CRIS records are derived, we anticipated that this linkage would facilitate the assessment of several pressing questions on the social determinants of onset, course and outcomes of severe mental health conditions that have thus far only been examined in case–control and prospective cohort studies limited by small sample sizes and significant attrition.

The purpose of this paper is to describe the creation of this data resource and to outline the methodology employed in linking individual records from the two sources. We also sought to describe the cohort's characteristics and to assess how these were associated with successful matches to census records. Finally, to evaluate the potential influence of records not matching on study outcomes, we compared unweighted and inverse probability weighted mortality estimates.

## METHODS
### Data sources used for creating the cohort
#### Clinical Record Interactive Search
SLaM provides mental healthcare to approximately 1.3 million residents in an urban, ethnically diverse and relatively deprived catchment area comprised of four South London boroughs: Croydon, Lambeth, Lewisham and Southwark. It is one of Europe's largest mental healthcare providers and covers all mental health services provided by the National Health Service (NHS), including

the improving access to psychological therapies (IAPT) service, child and adolescent mental health services and adult mental health, as well as general hospital liaison and various embedded specialist services (eg, the eating disorders outpatient service). Since 2007, clinical records for all SLaM services have been electronic-only, provided by its electronic patient journey system in the form of tick boxes, drop-down lists, free text and document attachments.[3 5] The CRIS application was developed to enable these records to be used for research within a robust data security and governance framework requiring a combination of data processing pipelines, including de-identification.[3] Thus, CRIS provides the entirety of a patient's mental health record, including information from structured data fields (eg, age, sex, diagnosis), but also de-identified free-text information, such as clinical correspondence letters, documents outlining care plans and detentions under the mental health act and routine clinical notes. A challenge with EHR systems is that some information may be captured poorly in structured fields and may instead be located in the clinical notes as free text, which is difficult to extract at scale. The CRIS platform enables the application of natural language processing (NLP) algorithms to convert unstructured text into relevant structured fields. These approaches have been successfully deployed previously to improve the identification of clinical diagnoses and symptoms, occupations and other important indicators of mental health.[3 6] Diagnostic data is captured through codes from the 10th edition of the International Classification of Disease (ICD), which may appear in both structured and unstructured data fields.

### 2011 census data
We used the results from the 2011 census of England and Wales as they were the most recent at the time that we initiated this data linkage project. The 2011 census was sent out to every household in England and Wales, and additional measures were taken to ensure the representation of individuals living in communal establishments, such as care homes, prisons and student halls, and of individuals without a fixed address, such as travellers or rough sleepers.[6] The person response rate for the 2011 census was 94%, making it the most comprehensive and representative source of socioeconomic and demographic data in England and Wales.[7] Census variables are categorised as 'standard' or 'derived', depending on whether the information they pertain to was explicitly referred to in census questions or derived from respondents' responses to other questions.[8] For an exhaustive list of variables and more information about the census, please see https://www.ons.gov.uk/census/2011census.

### Linked data set creation
We sought access to identifiable information for all individuals who had interacted with SLaM mental health services, including IAPT, up until 31 December 2018. This was done through the Health Research Authority

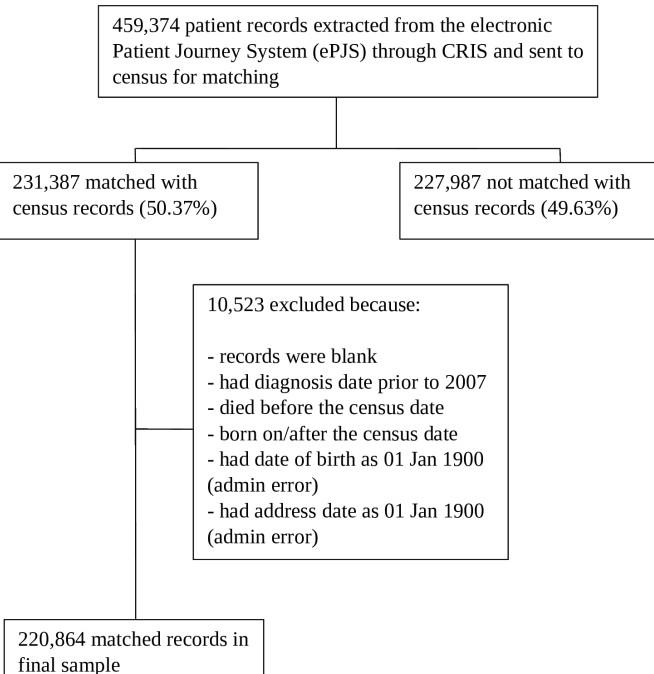

459,374 patient records extracted from the electronic Patient Journey System (ePJS) through CRIS and sent to census for matching

231,387 matched with census records (50.37%)

227,987 not matched with census records (49.63%)

10,523 excluded because:

- records were blank
- had diagnosis date prior to 2007
- died before the census date
- born on/after the census date
- had date of birth as 01 Jan 1900 (admin error)
- had address date as 01 Jan 1900 (admin error)

220,864 matched records in final sample

**Figure 1** Flow chart illustrating the sample selection process for the census matched/not matched data set. CRIS, Clinical Record Interactive Search.

by obtaining approval from the Confidential Advisory Group to identify patients under Section 251.[9] The reason for seeking access was to enable the linkage of records from CRIS and the 2011 census, which do not have a common identifier (eg, NHS number) and therefore must be linked through the use of identifiable information, such as name, date of birth and address. Records from CRIS were then supplied to the ONS, who acted as the trusted linkage function on behalf of the Administrative Data Research Centre for England and conducted the linkage to the 2011 census. Once records had been matched, identifiable information was removed and each of the records were given an identifier. The de-identified matched file was then hosted in the ONS secure environment, and accessible only to accredited researchers with project-specific approvals to access the data.

For the present analyses, we report associations between the clinical data set (CRIS) and the census match 'flag' generated following linkage. We removed observations if they contained erroneous birthdates (eg, year of birth was 1900), or if individuals had died before the census (23 March 2011) or were born afterwards (figure 1). Research Ethics Committee (REC) approvals for the establishment of the linked research database were also obtained, which was approved in addition to the existing REC approvals for CRIS (see Ethics approval section below).

### Linkage methodology

Records were linked deterministically through a series of matchkeys comprised of information common to both data sets to create unique identifiers. Because a single matchkey might be unable to resolve inconsistencies

between data sources, multiple matchkeys were employed. Table 1 summarises each matchkey, the degree to which they uniquely identified records in each data set, the proportion of CRIS to census matches and the specific discrepancy they intended to address. For instance, matchkey two did not include postcode, thereby allowing records to match on name and date of birth, even if the individual's residence had changed. Matchkeys were ranked by the proportion of unique observations that they identified and required exact matches on all the selected variables. To reduce the risk of false positives, records only linked on a matchkey if it was unique on both data sets. That is, when a record in one data set matched multiple records in the other data set, no matches were made and a new match was instead attempted with the next matchkey in the hierarchy. Once records matched, they were removed from the pool of records eligible to be selected for matching; another match with these records could therefore no longer be attempted. This means that there was no way to review and unlink matches made earlier in the hierarchy on the basis that the true match was identified at later stages of the matching procedure. Matchkeys 1–11 constitute a set of standard matchkeys that are routinely employed when data owned by the ONS is linked to another data set.[10] We also investigated whether the number of linked records could be increased by attempting further linkage with a set of experimental matchkeys on a randomly selected sample of CRIS data. This additional analysis resulted in matchkey 12.

### Measures

We examined an array of routinely recorded sociodemographic and clinical variables in the health record as predictors for successful matching (successful matching denoted through a 'match flag' as described above), including age, sex, ethnicity, marital status, referral date, history of admission to psychiatric hospital, clinical diagnosis by ICD-10 chapter and frequency of service contact. We determined frequency of contact with services by counting the number of times they had been referred. This information was primarily sourced from structured data fields in the health record (eg, a drop-down list). Diagnostic information was supplemented by meta-data derived from a bespoke validated NLP algorithm applied to text fields (eg, clinical correspondence).[3 11] We classified psychiatric disorder diagnoses according to ICD-10 F chapter headings, with an additional 'other diagnoses' category (eg, 'Unspecified mental disorder'). When patients had multiple diagnoses, we used the information in the 'primary diagnosis' field. We categorised ethnicity following the '18+1' ONS standard,[12] although we merged some categories due to low cell counts. Including an aggregation of all mixed ethnicity groups. Similarly, we placed individuals who were married or in a civil union in the same category. Age was calculated by subtracting the date of patients' first recorded contact with services from their birthdates and arranged into seven age bands (less than 25 years old, 25–34, 35–44, 45–54, 55–64, 65 years

**Table 1** Matchkey composition, uniqueness by data set and discrepancy addressed

| | | Uniqueness by data set % | | | |
|---|---|---|---|---|---|
| | Matchkey | Census | CRIS | CRIS to census match rate (N=231 387 (%)) | Issue addressed by matchkey |
| 1 | Forename, surname, DOB, sex, postcode | 100 | 98.7 | 87 780 (39.0) | None – exact agreement |
| 2 | Forename, surname, DOB, sex | 99.6 | 96.3 | 30 019 (13.0) | Moving out of area |
| 3 | Forename initial, surname initial, DOB, sex, postcode district | 99.9 | 97.2 | 43 587 (18.8) | Forename, surname and postcode discrepancy |
| 4 | Forename initial, DOB, sex, postcode | 99.97 | 98.3 | 9545 (4.1) | Surname discrepancy |
| 5 | Surname initial, DOB, sex, postcode | 99.9 | 97.8 | 5241 (2.3) | Forename discrepancy |
| 6 | Forename, surname, sex, postcode | 99.9 | 98.3 | 23 635 (10.2) | Date of birth discrepancy |
| 7 | Forename bigram*, surname bigram, DOB, sex, postcode area | 99.8 | 97.1 | 17 016 (7.4) | Name discrepancy and moving within area |
| 8 | Forename, surname, year of birth, sex, postcode district | 99.8 | 97.7 | 3073 (1.3) | Date of birth and moving within area |
| 9 | First middle name, surname, DOB, sex, postcode | 99.96 | 98.2 | 48 (0.0) | Forename and middle name transpositions |
| 10 | Second middle name, surname, DOB, sex, postcode | 99.96 | 98.1 | 12 (0.0) | Forename and second middle name transposition |
| 11 | Forename, surname, DOB, postcode | 100 | 98.7 | 902 (0.4) | Sex discrepancy |
| 12 | Forename bigram, surname bigram, postcode | 93.6 | 95.8 | 10 529 (4.6) | Name, sex and date of birth discrepancy |

*Bigram refers to the first two letters of the name.
CRIS, Clinical Research Interactive Search.

or older). We also extracted information on incident inpatient admission. Clinical records in CRIS also store information on death, which is obtained on a monthly basis from the NHS' 'Service User Death Report'.[13] We used this information to examine mortality as a secondary outcome in order to assess and adjust for potential biases introduced by non-matching. We also explored if outcomes varied by deprivation with the Index of Multiple Deprivation (IMD), an area-level composite measure of deprivation based on income, employment, crime, barriers to housing, health and disability, living environment and skills and training.[14] IMD scores are provided for small geographical areas that correspond to approximately 1500 individuals, known as a lower-layer super output area. Scores are assigned according to a patient's postcode that was on record closest to the census date, and placed in quartiles, with a higher score indicating higher levels of deprivation.

### Statistical methods

Using the census match flag, we compared linked and unlinked records to better understand which factors were associated with successful linkage between CRIS and census records. Because ORs fail to approximate relative risks when outcomes are common, we estimated prevalence ratios directly through a modified Poisson model with a robust variance estimator following methods outlined by Zou.[15] We opted for this method over a log-binomial modelling approach as it addresses the potential issue of model non-convergence.[15] We estimated crude and adjusted (sex, age and area-level deprivation) prevalence ratios (PR) indicating the association between demographic (eg, ethnicity) and clinical characteristics (eg, psychiatric diagnosis, history of admission) recorded in CRIS and the probability of matching to census records.

### Weighted analyses

A potential issue with linking data sets is that not all records will match, and that this might introduce bias if some parameters (eg, gender) are related to both matching status and outcomes of interest.[16] One way of mitigating the influence of biases due to non-matching is through inverse probability weighting (IPW). IPW weights each observation inversely to its probability of being matched so that those which are less likely to be matched receive higher weight.[17] Because we had near complete data in CRIS on gender, age and area-level deprivation, irrespective of matching status, we could assess and adjust for non-matching related to these characteristics by weighting the matched sample. We calculated the probability of matching through a logistic regression by entering match status as the outcome variable (ie, 1=matched; 0=did not match), with age group, gender and deprivation quartile as covariates. These probabilities were then converted into weights using the following formula, with P indicating the

probability of matching of the jth observation: $1-P_j$. We then estimated weighted and unweighted PR to measure the association between demographic (eg, marital status, ethnicity) and clinical variables (ie, diagnosis of a mental disorder, history of admission, frequency of contact with services) and all-cause mortality. The weighted and unweighted estimates were adjusted by age, gender and deprivation quartile.

### Patient and public involvement

Patient involvement was supported through consultation with the SLaM Clinical Data Linkage Service Data Linkage Service User and Carer Advisory Group, an advisory group of carers and individuals with lived experience of mental illnesses and mental health service use,[18] who were consulted at key points during the project. In addition, a CRIS oversight committee which is chaired by a service user, approves all projects proposing to use CRIS-linked data.

## RESULTS
### Cohort characteristics

We identified 459 374 records in CRIS, of which 231 387 (50.4%) matched the 2011 census through matchkeys 1–12 (table 1). We then applied further exclusion criteria, reducing our matched cohort to 220 864 cases (figure 1), which is the denominator for all proportions reported below. Just over half of total cohort members were women (54.6%) and the largest ethnic group was White British (52.9%), followed by Black Caribbean (13.8%) and Black African (4.8%). Nearly two-thirds (65.7%) of cohort members were single and/or separated. The average age of the cohort was 37 (SD: 20).

### Predictors of non-linkage

We observed differences within all demographic and clinical categories that we examined as predictors for matching success (table 2). For sex, men were less likely to match compared with women (PR 0.92, 95% CI 0.91 to 0.92). Relative to the youngest age group, those aged between 25 and 44 matched less frequently, but conversely, individuals 44 years or older were more likely to match, with the oldest age group (65+) having the highest probability of matching (PR 1.31, 95% CI 1.29 to 1.34). Widowed (PR 1.27, 95% CI 1.25 to 1.28) and married (PR 1.24, 95% CI 1.23 to 1.25) individuals matched more often than those whose who were unmarried. The probability of matching was lower for all minority ethnic groups compared with the White British group, with individuals identifying as White Other or Black African ethnicity the least likely to match. We observed a monotonic relationship between deprivation and matching success, with matching probability decreasing as deprivation increased. Matching success also appeared to vary by referral year, with the highest proportion (59.1%) seen in individuals referred in 2011 (the year of the census), with the next highest in the year after (2012; 57.9%) and before (2010; 55.9%)

(figure 2). Matching success varied by ICD-10 diagnosis (table 2), with relatively lower rates in individuals diagnosed with mental and behavioural disorders due to psychoactive substance use (F10–F19) or schizophrenia, schizotypal and delusional disorders (F20–F29) (PRs 0.86, 95% CI 0.85 to 0.87, and 0.91, 95% CI 0.89 to 0.92, respectively), and higher rates in those with organic mental disorders (F00–F09) (PR 1.38, 95% CI 1.36 to 1.40). Similarly, frequent contact with services was associated with a higher probability of matching (1–10 contacts: PR 1.04, 95% CI 1.04 to 1.05) compared with individuals without repeated contacts.

### Weighted versus unweighted mortality estimates

Weighted PR estimating risk of death tended to be higher for most categories examined compared with unweighted estimates (table 3); however, the differences were generally very small.

After adjusting for age, gender and deprivation quartile, individuals who were widowed were at the highest risk of death (table 3). Relative to other minority ethnic groups, the White British ethnic category was associated with the highest risk of death, as indicated by the lower PR in all other ethnic groups. However, weighted estimates for the association between ethnicity and all-cause mortality did not vary greatly, compared with unweighted estimates. As can be seen in table 3, all psychiatric disorders were associated with an increased risk of death, except for behavioural and emotional disorders with onset usually occurring in childhood and adolescence.

## DISCUSSION
### Summary of results

To our knowledge, this is the first time in which large-scale routine EHRs from a major secondary mental healthcare provider have been successfully linked to individual-level socio-demographic data from census in England. The resultant data set draws from an urban and ethnically diverse catchment area from which 220 864 secondary mental healthcare records were linked deterministically to detailed socio-demographic data from the 2011 census of England and Wales. Overall, half (50.4%) of records in the secondary mental healthcare data set linked to the 2011 census, and our analyses revealed differences between matched and non-matched records with respect to several socio-demographic and clinical characteristics. We observed the lowest match rates among young adults, individuals living in more deprived areas and among members of ethnic minority groups. We applied weights to assess how non-matching influenced mortality estimates and observed negligible differences between unweighted and weighted estimates, suggesting that non-linkage to census did not significantly bias associations.

### Analysis of records not matching

There are multiple reasons why non-linkage might occur. First, the match rate in our study will have been

**Table 2** Clinical Research Interactive Search cohort characteristics and their association with census matching

| Cohort characteristics<br>N=420 387 | Matched<br>N=220 864 (%) | Non-matched<br>N=199 523 (%) | Prevalence ratio<br>(95% CI) | Adjusted prevalence<br>ratio (95% CI)* |
|---|---|---|---|---|
| Gender | | | | |
| Female | 125 014 (56.6) | 104 008 (52.3) | Reference | Reference |
| Male | 95 669 (43.3) | 95 015 (47.7) | 0.92 (0.91 to 0.92) | 0.93 (0.92 to0.94) |
| Other | 16 (0.1) | 26 (0.1) | 0.70 (0.47 to 1.03) | 0.69 (0.46 to 1.04) |
| Marital status† | | | | |
| Single/separated | 86 472 (62.1) | 82 129 (70.0) | Reference | Reference |
| Cohabiting | 9519 (6.8) | 9628 (8.2) | 0.97 (0.95 to 0.98) | 0.98 (0.96 to 1.00) |
| Divorced | 5227 (3.8) | 4228 (3.6) | 1.07 (1.05 to 1.09) | 1.15 (1.13 to 1.17) |
| Married/civil union | 30 249 (21.7) | 17 139 (14.6) | 1.24 (1.23 to 1.25) | 1.01 (0.99 to 1.03) |
| Widowed | 7862 (5.6) | 4197 (3.6) | 1.27 (1.25 to 1.28) | 1.01 (0.99 to 1.03) |
| Age group | | | | |
| 24 and under | 76 826 (34.8) | 70 351 (35.4) | Reference | Reference |
| 25–34 | 38 248 (17.3) | 52 513 (26.4) | 0.81 (0.80 to 0.81) | 0.81 (0.81 to 0.82) |
| 35–44 | 35 197 (16.0) | 34 898 (17.5) | 0.96 (0.95 to 0.97) | 0.98 (0.97 to 0.99) |
| 45–54 | 29 481 (13.4) | 21 115 (10.6) | 1.12 (1.11 to 1.13) | 1.13 (1.12 to 1.14) |
| 55–64 | 15 837 (7.2) | 8440 (4.2) | 1.25 (1.24 to 1.26) | 1.25 (1.23 to1.28) |
| 65+ | 25 081 (11.4) | 11 685 (5.9) | 1.31 (1.30 to 1.32) | 1.27 (1.26 to 1.28) |
| Ethnicity | | | | |
| White British | 105 578 (60.5) | 68 008 (44.2) | Reference | Reference |
| Irish | 3086 (1.8) | 3435 (2.2) | 0.78 (0.76 to 0.80) | 0.79 (0.77 to 0.81) |
| Black Caribbean | 22 348 (12.8) | 23 023 (15.0) | 0.81 (0.80 to 0.82) | 0.84 (0.83 to 0.85) |
| Black African | 8420 (4.8) | 12 141 (7.9) | 0.67 (0.66 to 0.68) | 0.72 (0.71 to 0.73) |
| Indian | 3653 (2.1) | 2906 (1.9) | 0.92 (0.90 to 0.94) | 0.92 (0.90 to 0.94) |
| Pakistani | 1150 (0.7) | 1340 (0.9) | 0.76 (0.73 to 0.79) | 0.79 (0.76 to 0.82) |
| Bangladeshi | 721 (0.4) | 680 (0.4) | 0.85 (0.80 to 0.89) | 0.91 (0.86 to 0.96) |
| Chinese | 801 (0.5) | 1076 (0.7) | 0.70 (0.67 to 0.74) | 0.74 (0.70 to 0.78) |
| Other Asian | 3192 (1.8) | 4024 (2.6) | 0.73 (0.71 to 0.75) | 0.76 (0.74 to 0.78) |
| Other Ethnic | 9546 (5.5) | 12 002 (7.8) | 0.73 (0.72 to 0.74) | 0.77 (0.76 to 0.79) |
| Other White | 11 488 (6.6) | 20 046 (13.0) | 0.60 (0.59 to 0.61) | 0.65 (0.64 to 0.66) |
| Mixed, including other mixed | 4653 (2.7) | 5065 (3.3) | 0.79 (0.77 to 0.80) | 0.84 (0.82 to 0.86) |
| Deprivation quartile‡ | | | | |
| 1 (least deprived) | 62 673 (29.4) | 36 748 (20.1) | Reference | Reference |
| 2 | 51 957 (24.3) | 46 958 (25.7) | 0.83 (0.83 to 0.84) | 0.85 (0.83 to 0.86) |
| 3 | 50 214 (23.5) | 49 178 (26.9) | 0.80 (0.80 to 0.81) | 0.81 (0.80 to 0.82) |
| 4 (most deprived) | 48 634 (22.8) | 49 978 (27.3) | 0.78 (0.78 to 0.79) | 0.79 (0.78 to 0.80) |
| Any psychiatric diagnosis | | | | |
| No | 57 964 (26.2) | 61 632 (30.9) | Reference | Reference |
| Yes | 162 900 (73.8) | 137 891 (69.1) | 1.12 (1.11 to 1.12) | 1.09 (1.08 to 1.10) |
| Psychiatric diagnosis by International Classification of Disease-10 chapter | | | | |
| No record of diagnosis | 57 964 (26.2) | 61 632 (30.9) | Reference | Reference |
| F00–F09: Organic, including symptomatic, mental disorders | 13 133 (5.9) | 1.07 (1.05, 1.08) | 1.07 (1.05 to 1.08) | 1.07 (1.05 to 1.08) |

Continued

**Table 2** Continued

| Cohort characteristics N=420387 | Matched N=220864 (%) | Non-matched N=199523 (%) | Prevalence ratio (95% CI) | Adjusted prevalence ratio (95% CI)* |
|---|---|---|---|---|
| F10–F19: Mental and behavioural disorders due to psychoactive substance use | 10442 (4.7) | 0.89 (0.87, 0.90) | 0.89 (0.87 to 0.90) | 0.89 (0.87 to 0.90) |
| F20–F9: Schizophrenia, schizotypal and delusional disorders | 8363 (3.8) | 0.92 (0.91, 0.94) | 0.92 (0.91 to 0.94) | 0.92 (0.91 to 0.94) |
| F30–F36: Mood (affective) disorders | 44161 (20.0) | 1.10 (1.09, 1.11) | 1.10 (1.09 to 1.11) | 1.10 (1.09 to 1.11) |
| F40–F48: Neurotic, stress-related and somatoform disorders | 25854 (11.7) | 1.14 (1.13, 1.16) | 1.14 (1.13 to 1.16) | 1.14 (1.13 to 1.16) |
| F50–F59: Behavioural syndromes associated with physiological disturbances and physical factors | 4965 (2.2) | 1.27 (1.25, 1.29) | 1.27 (1.25 to 1.29) | 1.27 (1.25 to 1.29) |
| F60–F69: Disorders of adult personality and behaviour | 1312 (0.6) | 0.99 (0.95, 1.03) | 0.99 (0.95 to 1.03) | 0.99 (0.95 to 1.03) |
| F70–F79: Mental retardation | 640 (0.3) | 0.97 (0.92, 1.03) | 0.97 (0.92 to 1.03) | 0.97 (0.92 to 1.03) |
| F80–F89: Disorders of psychological development | 4545 (2.1) | 1.38 (1.36, 1.41) | 1.38 (1.36 to 1.41) | 1.38 (1.36 to 1.41) |
| F90–F98: Behavioural and emotional disorders with onset usually occurring in childhood and adolescence | 7060 (3.2) | 1.22 (1.20, 1.24) | 1.22 (1.20 to 1.24) | 1.22 (1.20 to 1.24) |
| F99: Unspecified mental disorder | 17611 (8.0) | 1.12 (1.11, 1.14) | 1.12 (1.11 to 1.14) | 1.12 (1.11 to 1.14) |
| Other diagnoses | 24814 (11.2) | 1.09 (1.08, 1.10) | 1.09 (1.08 to 1.10) | 1.09 (1.08 to 1.10) |
| History of admission | | | | |
| No | 210526 (95.3) | 187743 (94.1) | Reference | Reference |
| Yes | 10338 (4.7) | 11780 (5.9) | 0.88 (0.87 to 0.90) | 0.91 (0.90 to 0.93) |
| Face-to-face contacts | | | | |
| No contacts | 115430 (52.3) | 110632 (55.4) | Reference | Reference |
| 1–10 contacts | 67802 (30.7) | 59442 (29.8) | 1.04 (1.04 to 1.05) | 1.01 (1.01 to 1.02) |
| 11+ contacts | 37632 (17.0) | 29449 (14.8) | 1.10 (1.09 to 1.11) | 1.06 (1.06 to 1.07) |

*Prevalence ratios were adjusted for sex, age group and area level deprivation.
†The divorced and widowed categories also included civil unions that had ended, whether due to death or legal dissolution of the civil union.
‡Deprivation was measured through the Index of Multiple Deprivation.

inherently constrained by the proportion of cases in the CRIS cohort that responded to the 2011 census in the first place. The average response rate within the four London boroughs that comprise the SLaM catchment was lower (88%) compared with the national average (94%).[7] Among younger individuals (25–34 years old), who constituted a large proportion of our sample, the response rate was even lower in this region (84%). More mobile populations, which may include migrants and other groups temporarily moving into an area for work alongside people with severe mental illnesses,[19] may have been less likely to have taken part in the census. Individuals who moved into the SLaM catchment area and accessed services after 2011 would by default be unable to match. In addition, a growing body of evidence shows that racially minoritised groups, migrants and other socioeconomically marginalised groups are more likely to face discrimination in their interaction with governmental institutions in the UK, such as the police and the criminal justice system[20 21] and the NHS.[22] Previous studies have highlighted that Black and South Asian people may have concerns around how their data is safeguarded by institutions[23] and it is conceivable that this is manifested in lower rates of participation, although this could be explored in future work. Whatever the cause may be, it would nevertheless seem improbable that our match rate would exceed the average census response rate specific to the SLaM region or the various demographic groups that were prevalent in our sample. It is also well established that unit non-response can be considerable among individuals with a history of mental health disorders, who because of their illnesses might find it challenging to participate[24]

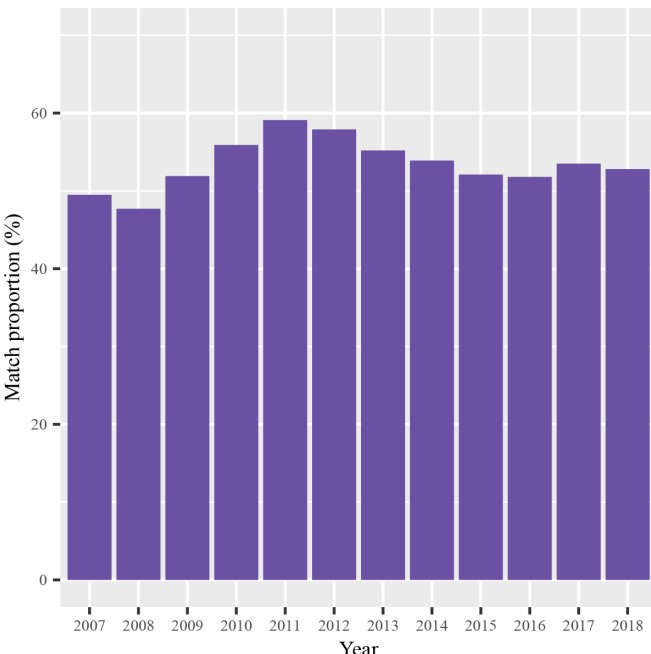

**Figure 2** Proportion of electronic patient records identified via the Clinical Research Interactive Search matched to census by referral year.

or may be more mobile.[19] Individuals with mental disorders are also more likely to experience objective social isolation (eg, have fewer measurable contacts with other individuals)[25] and might consequently be less likely to be captured through proxy responses (ie, family members responding in their stead). Indeed, surveys conducted annually since 2004 by the Quality Care Commission, the independent regulator of healthcare in the UK, have never observed response rates of above 41% in community mental health samples.[26]

Another factor that merits consideration is the underlying methodology employed in the matching itself. In our study, records were matched deterministically through matchkeys comprised of administrative information collected in both data sets. Inaccuracies or differences (eg, wrong postcode, incorrect date of birth, name changes due to marriage or alternative or erroneous spelling of names) in how these data were recorded might therefore have prevented some records from successfully matching. For example, previous linkage of health records to the census in Scotland highlighted a higher chance of clerical error with respect to the spelling of names for minority ethnic groups, leading to lower match rates.[27] As individuals from these groups were preponderant in our cohort, it is possible that clerical error accounted for a degree of non-matching in our study. Moreover, because most matchkeys required postcode information to match and because the match rate peaked among individuals who were referred the year the census was taken, it is possible that the deterministic matching methodology that we employed also missed some individuals who had a different address at the time they interacted with SLaM services and responded to the census. This is supported

by higher observed levels of matching (60%) for those with an address recorded in the mental health records at the time of census, in 2011, and is consistent with the interpretation that a high proportion of the sample in this study were potentially more mobile. Comparisons to previous efforts of linking the 2011 census to other administrative data could help disentangle the relative effects of sample-specific non-participation (eg, cohort member mobility or non-participation due to mental illness) and issues related to the methodology itself (eg, sensitivity of matchkeys). However, data linkage methods and the measurement of the linkage quality are continuously evolving within the ONS following the adaptation of new working environments and data sharing agreements, which preclude a fair comparison to other data linkage efforts involving the 2011 census. Our weighted analyses nevertheless indicated that missingness had a negligible influence on relevant study outcomes, such as associations of clinical/socio-demographic characteristics with all-cause mortality.

Finally, together with existing evidence from cohort studies of substantial attrition among participants diagnosed with mental illnesses, and of non-participation in community surveys, our findings point to non-response being a significant contributor to the low match-rate that we observed. Since the census informs the planning, funding and commissioning of local services, such as schools and health services, the potential under-representation of individuals with mental illnesses is concerning and merits further investigation.

### Strength and weaknesses
We believe that this is the first study to link census data in England to clinical records from a population in contact with secondary mental healthcare services. Because of the cohort's size, unique socio-demographic composition and abundant individual-level data on a multitude of important socio-demographic indicators provided by the linkage, we expect this data set to facilitate novel investigations into health inequalities among people living with mental disorders. For example, most prior research based on EHRs in the UK have relied on area level measures of socioeconomic status, such as the IMD, which itself is derived from census attributes.[14] Smith *et al*, By linking to clinical records to the census at the individual level, we could obtain a more accurate measure of the socioeconomic indicators. The overall size of the cohort is several magnitudes larger than previous UK-based mental health cohorts,[28] particularly with respect to ethnic minority groups and specific clinical subpopulations (eg, individuals with severe mental illnesses). The degree of non-linkage that we observed is a potential source of bias. However, we had comprehensive data on many relevant characteristics for the fully enumerated cohort, irrespective of matching status and could therefore determine through non-response weighting the relative influence that missingness related to these characteristics had, on all-cause mortality estimates. We intend to incorporate

**Table 3** Characteristics of census matched Clinical Research Interactive Search cases and unweighted and weighted prevalence ratios for all-cause mortality

| Cohort characteristics (N=220 864) | Deceased (N=18 363) | Alive (N=202 501) | Prevalence ratio (95% CI)* | |
| --- | --- | --- | --- | --- |
| | | | Unweighted | Weighted |
| Marital status† | | | | |
| Single/separated | 5078 (32.8) | 81 394 (65.7) | Reference | Reference |
| Cohabiting | 147 (1.0) | 9372 (7.6) | 0.44 (0.38 to 0.51) | 0.41 (0.35 to 0.48) |
| Divorced | 891 (5.8) | 4336 (3.5) | 0.83 (0.78 to 0.88) | 0.82 (0.77 to 0.87) |
| Married | 5140 (33.2) | 25 109 (20.3) | 0.85 (0.82 to 0.88) | 0.83 (0.81 to 0.87) |
| Widowed | 4203 (27.2) | 3659 (3.0) | 1.15 (1.12 to 1.19) | 1.14 (1.11 to 1.18 |
| Ethnicity | | | | |
| White British | 12 033 (73.3) | 93 545 (59.1) | Reference | Reference |
| Irish | 626 (3.8) | 2460 (1.6) | 0.99 (0.93 to 1.05) | 0.99 (0.93 to 1.06) |
| Black Caribbean | 1322 (8.1) | 21 026 (13.3) | 0.72 (0.69 to 0.75) | 0.72 (0.68 to 0.75) |
| Black African | 316 (1.9) | 8104 (5.1) | 0.62 (0.56 to 0.69) | 0.63 (0.57 to 0.70) |
| Indian | 360 (2.2) | 3293 (2.1) | 0.81 (0.74 to 0.88) | 0.80 (0.74 to 0.87) |
| Pakistani | 76 (0.5) | 1074 (0.7) | 0.78 (0.64 to 0.94) | 0.80 (0.65 to 0.98) |
| Bangladeshi | 23 (0.1) | 698 (0.4) | 0.56 (0.39 to 0.81) | 0.57 (0.39 to 0.83) |
| Chinese | 46 (0.3) | 755 (0.5) | 0.71 (0.56 to 0.89) | 0.71 (0.56 to 0.90) |
| Other Asian | 225 (1.4) | 2967 (1.9) | 0.74 (0.66 to 0.83) | 0.75 (0.67 to 0.84) |
| Other Ethnic | 551 (3.4) | 8995 (5.7) | 0.83 (0.77 to 0.90) | 0.82 (0.76 to 0.89) |
| Other White | 801 (4.9) | 10 687 (6.8) | 0.80 (0.76 to 0.85) | 0.80 (0.75 to 0.85) |
| Mixed, including other mixed | 43 (0.3) | 4610 (2.9) | 0.32 (0.24 to 0.42) | 0.30 (0.22 to 0.40) |
| Any psychiatric diagnosis | | | | |
| No | 2980 (9.8) | 116 616 (29.9) | Reference | Reference |
| Yes | 27 407 (90.2) | 273 384 (70.1) | 2.66 (2.52 to 2.80) | 2.70 (2.56 to 2.84) |
| Psychiatric diagnosis by International Classification of Disease-10 chapter | | | | |
| No record of diagnosis | 2980 (9.8) | 116 616 (29.9) | Reference | Reference |
| F00–F09l: Organic, including symptomatic, mental disorders | 10 924 (35.9) | 8723 (2.2) | 3.25 (3.08 to 3.43) | 3.32 (3.14 to 3.51) |
| F10–F19: Mental and behavioural disorders due to psychoactive substance use | 2784 (9.2) | 22 233 (5.7) | 4.47 (4.16 to 4.81) | 4.77 (4.43 to 5.13) |
| F20–F9: Schizophrenia, schizotypal and delusional disorders | 1889 (6.2) | 17 099 (4.4) | 2.88 (2.66 to 3.11) | 3.05 (2.81 to 3.31) |
| F30–F36: Mood (affective) disorders | 4607 (15.2) | 76 513 (19.6) | 2.23 (2.10 to 2.36) | 2.23 (2.10 to 2.37) |
| F40–F48: Neurotic, stress-related and somatoform disorders | 1288 (4.2) | 45 145 (11.6) | 1.58 (1.47 to 1.70) | 1.54 (1.43 to 1.66) |
| F50–F59: Behavioural syndromes associated with physiological disturbances and physical factors | 123 (0.4) | 7831 (2.0) | 1.51 (1.22 to 1. 85) | 1.54 (1.24 to 1.90) |
| F60–F69: Disorders of adult personality and behaviour | 149 (0.5) | 2668 (0.7) | 3.30 (2.66 to 4.10) | 3.57 (2.85 to 4.48) |
| F70–F79: Mental retardation | 137 (0.5) | 1177 (0.3) | 4.30 (3.35 to 5.53) | 4.53 (3.49 to 5.87) |
| F80–F89: Disorders of psychological development | 60 (0.2) | 6783 (1.7) | 1.40 (1.01 to 1.95) | 1.33 (0.95 to 1.87) |
| F90–F98: Behavioural and emotional disorders with onset usually occurring in childhood and adolescence | 53 (0.2) | 12 099 (3.1) | 0.85 (0.59 to 1.24) | 0.88 (0.60 to 1.29) |
| F99: Unspecified mental disorder | 1869 (6.2) | 30 260 (7.8) | 2.61 (2.44 to 2.79) | 2.66 (2.48 to 2.85) |
| Other diagnoses | 3524 (11.6) | 42 853 (11.0) | 2.50 (2.35 to 2.65) | 2.55 (2.40 to 2.72) |

**Table 3** Continued

| Cohort characteristics (N=220 864) | Deceased (N=18 363) | Alive (N=202 501) | Prevalence ratio (95% CI)* | |
|---|---|---|---|---|
| | | | Unweighted | Weighted |
| History of admission | | | | |
| No | 17 207 (93.7) | 193 319 (95.5) | Reference | Reference |
| Yes | 1156 (6.3) | 9182 (4.5) | 1.43 (1.36 to 1.50) | 1.49 (1.42 to 1.57) |
| Face-to-face contacts | | | | |
| No contacts | 3465 (18.9) | 111 965 (55.3) | Reference | Reference |
| 1–10 contacts | 10316 (56.2) | 57 486 (28.4) | 2.42 (2.34 to 2.51) | 2.52 (2.42 to 2.62) |
| 11+ contacts | 4582 (25.0) | 33 050 (16.3) | 2.56 (2.47 to 2.67) | 2.68 (2.57 to 2.79) |

*All models adjusted for age, sex and deprivation quartile.
†Civil unions were also included in the divorced, married and widowed categories.

these weights in all future analyses to minimise sources of bias. Although the area is ethnically diverse with a good overall representation of Black Caribbean and Black African people, other prevalent ethnic minority groups in England, such as Indian, Pakistani and Bangladeshi populations, are less well represented. In addition, some characteristics that we examined as predictors for matching, such as ethnicity and marriage status, are inherently dynamic, which may have resulted in less precise estimates. Although the highly urban nature of the South London catchment area may be generalisable to other urbanised locations in England, inferences relating to more rural areas may not be possible. There is some evidence that matching of administrative records can be improved through the use of probabilistic techniques,[29] but these were not used by the ONS for this linkage. It is possible that we could have obtained a higher match rate had record matching been supplemented with probabilistic methods. Salary information, a direct measure of socioeconomic standing, is not collected in the census. However, it does contain data on numerous other factors which can be used to estimate individual wealth, including employment status, tenure, house composition and car ownership. One of the challenges with the linkage methods employed here is that we could not conclusively determine the exact causes of non-linkage. For instance, we could not quantify the relative degree to which non-linkage was caused by unit non-response or clerical errors in how data was recorded. Our study described the process of linking census to mental health electronic records. In the future, we plan to undertake assessments for the association of social and economic indicators from census with potential mental health outcomes. However, a limitation of census is that it is self-report, and this may lead to under-reporting for some important indicators (eg, migration status, employment status). This will need to be considered in future work. Finally, we could not examine cause-specific mortality, but will explore this in future analyses with linked data from the ONS mortality registration.

**Acknowledgements** We are grateful to Hitesh Shetty (SLAM-BRC CRIS team) for his support with data management.

**Contributors** JD-M conceived the study, designed the work and led acquisition of the linked data set and interpretation of findings. LC led design, analysis and interpretation of findings. JD-M and LC led drafting of the manuscript. AJ supported the design and acquisition of the linked data set, with MP. NC conducted the initial analysis of findings. MD advised on statistical analyses and interpretation. CM, MH, RHu, RHi, MW and RS contributed to the interpretation of findings. All authors were involved in drafting the work or revising it critically prior to submission and all authors approved the final version to be published and agree to be accountable for all aspects of the work. JD-M is the guarantor of the study.

**Funding** This paper represents independent research part-funded by the National Institute for Health Research (NIHR) Biomedical Research Centre at South London and Maudsley NHS Foundation Trust and King's College London. LC and MW are supported by a grant from the ESRC (ES/S002715/1). RHi is currently funded by a doctoral studentship granted by the UKRI ESRC LISS-DTP managed by King's College London. JD-M and CM are part supported by the ESRC Centre for Society and Mental Health at King's College London (ESRC Reference: ES/S012567/1) and by the National Institute for Health Research (NIHR) Biomedical Research Centre at South London and Maudsley NHS Foundation Trust and King's College London and the National Institute for Health Research (NIHR) Applied Research Collaboration South London (NIHR ARC South London) at King's College Hospital NHS Foundation Trust. MH is a NIHR Senior Investigator. RS is part-funded by: (1) the National Institute for Health Research (NIHR) Maudsley Biomedical Research Centre at the South London and Maudsley NHS Foundation Trust and King's College London; (2) the NIHR Applied Research Collaboration South London (NIHR ARC South London) at King's College Hospital NHS Foundation Trust; (3) UKRI – Medical Research Council through the DATAMIND HDR UK Mental Health Data Hub (MRC reference: MR/W014386); (4) the UK Prevention Research Partnership (Violence, Health and Society; MR-VO49879/1), an initiative funded by UK Research and Innovation Councils, the Department of Health and Social Care (England) and the UK devolved administrations and leading health research charities.

**Disclaimer** The views expressed are those of the authors and not necessarily those of the ESRC, NHS, the NIHR or the Department of Health and Social Care or King's College London.

**Competing interests** MH is principal investigator of the RADAR-CNS, a precompetitive public–private collaboration on mobile health funded by the Innovative Medicine Initiative with cash and in-kind contributions paid to the university from Janssen, Lundbeck, UCB, MSD and Biogen. RS declares research support in the last 3 years from Janssen, GSK and Takeda.

**Patient and public involvement** Patients and/or the public were involved in the design, or conduct, or reporting, or dissemination plans of this research. Refer to the Methods section for further details.

**Patient consent for publication** Not applicable.

**Ethics approval** CRIS has Research Ethics Committee approval as a source of anonymised data for secondary analysis (Oxford REC C, reference 18/SC/0372). The current CRIS-Census linkage was supported through: REC reference for CRIS-Census Linkage: 18/SC/0003. Additional approvals from the Confidential Advisory Group to access patient information without consent, for the purposes of linkage, were obtained (CAG S251 reference: 17/CAG/0204). Approvals were also sought and obtained from the National Statistician's Data Ethics Advisory Committee (NSDEC) for approvals to use linked CRIS-census data for specified projects.

**Provenance and peer review** Not commissioned; externally peer reviewed.

**Data availability statement** Data may be obtained from a third party and are not publicly available. Data are owned by a third party SLaM BRC CRIS tool which provides access to anonymised data derived from SLaM electronic medical records. These data can only be accessed by permitted individuals from within a secure firewall (ie, remote access is not possible and the data cannot be sent elsewhere) in the same manner as the authors.

**Open access** This is an open access article distributed in accordance with the Creative Commons Attribution 4.0 Unported (CC BY 4.0) license, which permits others to copy, redistribute, remix, transform and build upon this work for any purpose, provided the original work is properly cited, a link to the licence is given, and indication of whether changes were made. See: https://creativecommons.org/licenses/by/4.0/.

**ORCID iDs**
Lukasz Cybulski http://orcid.org/0000-0002-1774-7288
Natasha Chilman http://orcid.org/0000-0002-9661-5098
Jayati Das-Munshi http://orcid.org/0000-0002-3913-6859

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
