## [Reviewer comments · BMJ Open]

ARTICLE DETAILS

TITLE (PROVISIONAL)	Improving our understanding of the social determinants of mental health: A data linkage study of mental health records and the 2011 UK census.
AUTHORS	Cybulski, Lukasz; Chilman, Natasha; Jewell, Amelia; Dewey, Michael; Hildersley, Rosanna; Morgan, Craig; Huck, Rachel; Hotopf, Matthew; Stewart, Robert; Pritchard, Megan; Wuerth, Milena; Das-Munshi, Jayati

VERSION 1 – REVIEW

REVIEWER	Andres Roman-Urrestarazu University of Cambridge
REVIEW RETURNED	10-May-2023

GENERAL COMMENTS	Interesting paper and work. I have some serious concerns that hamper my initial enthusiasm for the authors work. 1) Why did they authors use census data? Is the assumption that this represents socioeconomic status better than other data sources? The main issue with this is the self reported nature of the source for data linkage. Some authors have posted concerns about this as this might misrepresent actual deprivation and therefore have used for example other measures of actual services received using educational or social care data. I would have liked a better discussion of the sources of bias beyond why some have used area measures of social deprivation etc. 2) The authors say the census is the most representative source for demographic and socio-economic data available. I would dispute this. In terms of demography they are right but there are a series of issues with the census that should be discussed such as for example no reporting of ethnicity below certain areas such as LSOA and also the issues around income and wealth that also are not captured in the depth and detailed required. I also think that there are other resources such as the Labour Force Survey and other that tend to capture better employment status and so forth. I would try and moderate such statements and try and take a less editorial approach to data source description. 3) In their explanation of direct and derived variables I would mention how these are not derived in other surveys eg LFS and would actually add this to a supplementary materials section as I do not believe this is something that adds any value and is openly available information. It reads 'wordy' as well. 4) We also use CRIS where I work and this does not come with certain important caveats that ideally should be presented. I think for example in the NLP discussion it would be great to mention methods
--

	used such as the 'labelling' of variables and more important how data resources depends also on data input. For example across NHS services there are concerns around diagnostic inputs and missingness. I would recommend the authors to discuss this in detail in the methods and also explain this so that other could understand and ideally be able to replicate the findings. As it stands CRIS looks like a black box and this hampers replicability of any findings. 5) If ONS did the linkage why not use for example the National Pupil Database that has information on claimed eligibility in the free school meals programme? 6) Because of the nature of the matching process, no replicability is possible byMa third party researchers. 7) Many of the match flags the authors included are dynamic. For example marriage status, and in my experience ethnicity. For example one person can identify as mixed race, black or mixed other in different iterations of data coding and even census. I would have liked a discussion around this crucial issue. 8) How stable are diagnostic categories in your trust? For example do you have cases of FEP, then Schizoaffective disorder and finally schizophrenia? Or maybe BED, then bulimia and finally anorexia? I think this is another dynamic issue around diagnostic accuracy in NHS services where payments are not linked to diagnosis (eg think DRG in the USA). 9) A poisson model seems adequate and deals with non-convergence. I would have liked a discussion though about multilevel modelling and maybe looking at mixed effects. In other words for example do health outcomes vary by area but also by provider even across the trust? At least I would have liked a discussion about this. 10) Why did the authors use crude prevalence estimates and not standardised across age and sex? Seems odd considering they are linking census data. Also it is important to standardise across ethnicity considering for example higher incidence of FEP in African Caribbean communities. Please justify why this was not done. 11) SLaM is located in a very particular city: London. The non match is important because of the transient nature of life in London with an important immigration flux and movement between people that come in and also that move away. How did the authors match that in its census linkage? I think there should be an exploration of this in the low matching obtained between their EHR data and the census data. 12) The authors mention distrust in service and institutions without evidence this is the case. I would avoid statements that are not supported in evidence. 13) I might have missed but I would have liked to see if their model was better or not than just assigning IMD as the main outcome for social deprivation? With the level of matching they discussed I would have liked a balanced discussion around this. I hope my comments are helpful in improving the quality of this necessary work.
--	---

REVIEWER	Foteini Tseliou Cardiff University
REVIEW RETURNED	03-Jul-2023

GENERAL COMMENTS	This is a well-conducted study that aimed to improve understanding of the social determinants of mental health using a record-linkage methodology. However, I have some comments on the methods implemented and how these could affect the interpretation of the observed results.
--

	Major comments 1. My primary comment is on the innovation of this paper. The methods used as sound and suitable, but I am not sure on what this analysis can add to the literature as similar linkages take place via the NISRA and the Northern Ireland Longitudinal and Mortality Study, the Scottish Longitudinal Study in Scotland as well as the Secure Anonymised Information Linkage Databank in Wales. Perhaps, if clearly research question was set out e.g. investigating social determinants of mental health as suggested in the title then it would make the paper stronger. 2. The title and abstract state that the focus is on social determinants, but the characteristics provided are primarily demographic e.g. gender and ethnicity, with the exception of deprivation quartile. If that was the focus of the paper, then additional factors such as home and car ownership, proximity to services as well as measures of employment or education status should be considered. 3. It is suggested that males were less likely to be linked than women. Usually, the opposite is the case due to change of name following marriage etc. Was there any additional analysis undertaken to double-check this finding? 4. Perhaps the addition of GP records where more up-to-date information on address is available could improve linkage success rates. Would that be available through the electronic healthcare records? 5. There was no major difference when accounting for face-to-face contacts. This is perhaps because the severity of condition might vary significantly. Perhaps sensitivity analyses exploring the reason/type of visit could provide further details. 6. It would also be interesting to control for death by suicide, if the cause of death is available as differences will be expected in comparison to all-cause mortality. Minor comments 7. Please clarify what “service contact” might involve. Currently it is not clear to the reader. 8. In the case of multiple diagnoses, how was the primary diagnosis chosen or were all diagnoses considered together?
--	---

VERSION 1 – AUTHOR RESPONSE

	Comment	Response
	Reviewer 1	
1	Why did they authors use census data? Is the assumption that this represents socioeconomic status better than other data sources? The main issue with this is the self reported nature of the source for data linkage. Some authors have posted concerns about this as this might misrepresent actual deprivation and therefore	We linked clinical records in CRIS to the census because clinical records lack the sociodemographic data that is in the census. We make this point in the introduction on page 4: “Data on occupational classification, long-term unemployment, ethnicity, housing tenure, education, migration, and other relevant socioeconomic measures are often either missing, inaccurate, or collected infrequently, hindering efforts to better understand relationships between mental health and socioeconomic and sociodemographic factors.”

have used for example other measures of actual services received using educational or social care data. I would have liked a better discussion of the sources of bias beyond why some have used area measures of social deprivation etc.	Most UK-based studies that utilise routinely collected clinical data (e.g., the Clinical Practice Research Datalink, Hospital Episode Statistics, etc.) have to date relied on the index of multiple deprivation (IMD) as the only measure of socioeconomic deprivation. Because the IMD is a composite measure that itself is derived from census measures, we can obtain a more accurate representation of deprivation than relying solely on the IMD. We have clarified this point in the in the discussion on page 12: “For example, most prior research based on EHRs in the UK have relied on area level measures of socioeconomic status, such as the IMD, which itself is derived from census attributes (Smith et al., 2015). By linking clinical records to the census at the individual level, we could obtain a more accurate measure of the socioeconomic indicators.” The Goldthorpe schema for occupational social class is a self-report measure of occupational social class which is a key indicator of socioeconomic status. It is widely accepted internationally and has been thoroughly validated. The National Statistics Socioeconomic Classification schema (NS-SEC) is based on the Goldthorpe schema and is collected through Census. See this page for further details: https://www.ons.gov.uk/methodology/classificationsandstandards/otherclassifications/thenationalstatistics socioeconomicclassificationnssecbasedonsoc2010 We believe this is one of the best measures for assessment of occupational social class as an indicator of socioeconomic status. This is also evidenced through its use to highlight health inequalities in landmark reports, such as the Marmot Review. References: Marmot, M. (2013). Fair society, healthy lives. Fair society, healthy lives, 1-74. Smith T, Noble M, Noble S, Wright G, McLennan D, Plunkett E. The English Indices of Deprivation 2015. Department for Communities and Local Government; 2015.
2 The authors say the census	The census is widely considered to be the gold standard for

a	is the most representative source for demographic and socio-economic data available. I would dispute this. In terms of demography they are right but there are a series of issues with the census that should be discussed such as for example no reporting of ethnicity below certain areas such as LSOA and also the issues around income and wealth that also are not captured in the depth and detailed required.	sociodemographic data in the UK and is for this reason used to inform the planning of public services in the UK, including education, transport, and healthcare. In this study we did not use census data which is publicly available. It is true that this data only is available at minimum LSOA level in order to maintain confidentiality. However, the innovation in our work is that we linked to individual-level census data. Therefore, all information sourced from census, including ethnicity, is at the individual level. We acknowledge that salary information isn't collected in the census, and mention this as a limitation on page 12: “Salary information, a direct measure of socioeconomic standing, is not collected in the census. However, it does contain data on numerous other factors which can be used to estimate individual wealth, including employment status, tenure, house composition and car ownership”
2 b	I also think that there are other resources such as the Labour Force Survey and other that tend to capture better employment status and so forth. I would try and moderate such statements and try and take a less editorial approach to data source description.	Whilst the Labour Force Survey (LFS) may have more detailed information on wealth and income than the census, it wouldn't be feasible to link it to individual mental health records in CRIS using the methods outlined in the paper. Firstly, each wave of the LFS is limited to a sample of the population (c.a. 36,000 households) and it is therefore likely that many individuals in the SLAM catchment area have not been approached to participate. The census is on the other hand sent out to all households in the UK. Second, the response rate to the LFS is much lower (c.a. 40-50% at best) than the census (c.a. 94%), which a) raises questions about the representativeness of this data, and b) would result in a very low match rate if similar response rates were obtained in our sample. Moreover, we suspect that the response rate to surveys like the census or the LFS is likely to be lower among individuals with severe mental disorders, which would further compound these issues. Finally, the census includes information on many other variables that are of interest to mental health researchers, such as migration status, and health and disability measures. Please see the links for figures on the LFS response rate: https://www.ons.gov.uk/employmentandlabourmarket/peopleinwork/employmentandemployeetypes/methodologies/labourforcesurveyperformanceandqualitymonitoringreports/labourforcesurveyperformanceandqualitymonitoringreportapriltojune2021#:~:text=the%20total%20response%20rate%20for,respectively%20in%20the%20previous%20quarter

		https://www.hse.gov.uk/statistics/lfs/about.htm
3	In their explanation of direct and derived variables I would mention how these are not derived in other surveys eg LFS and would actually add this to a supplementary materials section as I do not believe this is something that adds any value and is openly available information. It reads 'wordy' as well.	We have removed the sentence from the methods section under the '2011 census data' heading on page 5 that lists some of the standard and derived variables. We now direct the reader to the ONS website that contains an exhaustive list of all standard/derived variables: "For an exhaustive list of variables and more information about the census, please see https://www.ons.gov.uk/census/2011census ."
4	We also use CRIS where I work and this does not come with certain important caveats that ideally should be presented. I think for example in the NLP discussion it would be great to mention methods used such as the 'labelling' of variables and more important how data resources depends also on data input. For example across NHS services there are concerns around diagnostic inputs and missingness. I would recommend the authors to discuss this in detail in the methods and also explain this so that other could understand and ideally be able to replicate the findings. As it stands CRIS looks like a black box and this hampers replicability of any findings.	We have added some additional information about this and direct interested readers to two publications which discuss this issue in detail, on page 5: "A challenge with electronic health record systems is that some information may be captured poorly in structured fields and may instead be located in the clinical notes as free text, which is difficult to extract at scale. The CRIS platform enables the application of natural language processing (NLP) algorithms to convert unstructured text into relevant structured fields. These approaches have been successfully deployed previously to improve the identification of clinical diagnoses and symptoms, occupations, and other important indicators of mental health (3,6) "

5	If ONS did the linkage why not use for example the National Pupil Database that has information on claimed eligibility in the free school meals programme?	With this data linkage, we wanted to create a dataset for the study of adults with severe mental disorders. Linking to the National Pupil Database (NPD) would present its own set of issues as many individuals in the cohort will have gone to school before the establishment of the NPD. In addition, many residents in the SLAM catchment area will have migrated to the UK as adults, which means they also wouldn't have a record in the NPD. Data from CRIS has already been linked to the National Pupil Database for other studies on child mental health. Please see Downs et al., 2019 for more information. Reference: Downs, J. M., Ford, T., Stewart, R., Epstein, S., Shetty, H., Little, R., ... & Hayes, R. (2019). An approach to linking education, social care and electronic health records for children and young people in South London: a linkage study of child and adolescent mental health service data. BMJ open, 9(1), e024355.
6	Because of the nature of the matching process, no replicability is possible by third party researchers.	The linkage process was undertaken by ONS using validated methods that they have been used previously for other linkage studies, and which have been widely published to ensure replicability. The deterministic methods used in linking records have been described in detail in the report "Beyond 2011: Matching Anonymous Data" which we cite in the 'Linkage methodology' section on page 6. Reference: Office of National Statistics. Beyond 2011: Matching Anonymous Data. 2013.
7	Many of the match flags the authors included are dynamic. For example marriage status, and in my experience ethnicity. For example one person can identify as mixed race, black or mixed other in different iterations of data coding and even census. I would have liked a discussion around this crucial issue.	We now point out that this may have made our estimates less precise in the 'Strength and limitations' section on page 12: "In addition, some characteristics that we examined as predictors for matching, such as ethnicity and marriage status, are inherently dynamic, which may have resulted in less precise estimates."
8	How stable are diagnostic categories in your trust? For	Changes to a patient's diagnosis reflect clinical reality and are not necessarily an issue of accuracy. For example, as the reviewer

	example do you have cases of FEP, then Schizoaffective disorder and finally schizophrenia? Or maybe BED, then bulimia and finally anorexia? I think this is another dynamic issue around diagnostic accuracy in NHS services where payments are not linked to diagnosis (eg think DRG in the USA).	highlights, patient’s clinical presentations often change over time with corresponding changes to their diagnosis. These are captured in the clinical records but for the purposes of the present analyses we focused on those captured in the patient’s “primary diagnosis” field. “When patients had multiple diagnoses, we used the information in the ‘primary diagnosis’ field.” Page. 7
9	A poisson model seems adequate and deals with non-convergence. I would have liked a discussion though about multilevel modelling and maybe looking at mixed effects. In other words for example do health outcomes vary by area but also by provider even across the trust? At least I would have liked a discussion about this.	We have linked the data described in this paper to the ONS mortality registration as well and will describe this linkage in a separate publication. It is our intention to study mortality outcomes in future work using this data using mixed effects approaches and is beyond the scope of the current analysis. The current study’s aim was to describe the linkage process of CRIS to census and methods for mitigating bias due to non-matching.
10	Why did the authors use crude prevalence estimates and not standardised across age and sex? Seems odd considering they are linking census data. Also it is important to standardise across ethnicity considering for example higher incidence of FEP in African Caribbean communities. Please justify why this was not done.	We have added a column to table 2 with prevalence ratios adjusted for sex, age, and area level deprivation. We have made also made changes to the ‘Statistical methods’ section on page 8 to reflect this: “We estimated crude and adjusted (sex, age, and area-level deprivation) prevalence ratios (PR) indicating the association between demographic (e.g., ethnicity) and clinical characteristics (e.g., psychiatric diagnosis, history of admission) recorded in CRIS and the probability of matching to census records”
11	SLaM is located in a very particular city: London. The non match is important because of the transient nature of life in London with an important immigration flux and movement between people that come in and also that move away. How did the authors match that in its census linkage? I	The matchkeys used to link census records to CRIS were developed to account for potential discrepancies in information provided in CRIS and census, including information about an individual’s postcode. We discuss the issue of patient mobility and its potential influence on the observed match rate: “Moreover, because most matchkeys required postcode information to match and because the match rate peaked among individuals who were referred the year the census was taken, it is possible that

	think there should be an exploration of this in the low matching obtained between their EHR data and the census data.	the deterministic matching methodology that we employed also missed some individuals who had a different address at the time they interacted with SLAM services and responded to the census.” Page. 11
1 2	The authors mention distrust in service and institutions without evidence this is the case. I would avoid statements that are not supported in evidence.	We have added a citation from a report that highlighted low levels of trust among individuals from South Asian and Black communities with respect to sharing sensitive data, and amended the text in the discussion as follows: “Previous studies have highlighted that Black and South Asian people may have concerns around how their data is safeguarded by institutions (23) and it is conceivable that this is manifested in lower rates of participation, although this could be explored in future work” page. 10 Reference: Bailey-Wilson B, Wilkinson-Salamea C, Raidos D, Twins B, Imafidon K, McGarry N, et al. Diverse Voices on Data 2022 [Available from: https://understandingpatientdata.org.uk/sites/default/files/2022-04/Diverse%20voices%20on%20Data%20-%20Main%20report_0.pdf .
1 3	I might have missed but I would have liked to see if their model was better or not than just assigning IMD as the main outcome for social deprivation? With the level of matching they discussed I would have liked a balanced discussion around this.	Our outcomes were matching success and mortality. IMD was not an outcome. We examined IMD as a predictor for matching success. In our mortality analyses, the purpose was to compare weighted vs. unweighted estimates to gauge the influence of non-matching on mortality.
	Reviewer 2	
1	My primary comment is on the innovation of this paper. The methods used as sound and suitable, but I am not sure on what this analysis can add to the literature as similar linkages take place via the NISRA and the Northern Ireland	We are aware of the data linkages you describe, but there are some important differences. Firstly, the characteristics of the Northern Ireland Longitudinal Mortality Study and the Scottish Longitudinal Study are in terms of geography and demography different to the cohort that we describe in this paper. In contrast, the SLAM catchment area is highly urban and has proportionally more residents born outside the UK and from racialised minorities. We also linked records from IAPT and secondary mental health services, whereas the two other datasets rely on hospital records

	Longitudinal and Mortality Study, the Scottish Longitudinal Study in Scotland as well as the Secure Anonymised Information Linkage Databank in Wales. Perhaps, if clearly research question was set out e.g. investigating social determinants of mental health as suggested in the title then it would make the paper stronger.	only. The SLAM records also enable free text searches of deidentified clinical records as we have highlighted in the paper and above, which is further advantage over the other linkages. Thus, the CRIS-census linkage provides a unique perspective on mental health inequalities which is qualitatively different from the other, ultimately complementary data resources.
2	The title and abstract state that the focus is on social determinants, but the characteristics provided are primarily demographic e.g. gender and ethnicity, with the exception of deprivation quartile. If that was the focus of the paper, then additional factors such as home and car ownership, proximity to services as well as measures of employment or education status should be considered.	This study describes the methodology used in linking clinical records to the 2011 census and explores the association between demographic characteristics and matching success. We outline in the introduction the rationale for conducting this linkage, i.e., the lack of detailed sociodemographic information in clinical records: “However, despite their strengths, EHRs typically contain limited information on socioeconomic characteristics at the individual level. Data on occupational classification, long-term unemployment, ethnicity, housing tenure, education, migration, and other relevant socioeconomic measures are often either missing, inaccurate, or collected infrequently, hindering efforts to better understand relationships between mental health and socioeconomic and sociodemographic factors.” Page 4. At the end of the introduction, we state the aims of the study: “The purpose of this paper is to describe the creation of this data resource and to outline the methodology employed in linking individual records from the two sources. We also sought to describe the cohort’s characteristics and to assess how these were associated with successful matches to census records. Finally, to evaluate the potential influence of records not matching on study outcomes, we compared unweighted and inverse probability weighted mortality estimates” Page 4. We believe that our title and abstract reflects these key aims (emphasis in bold).

		Title: "Improving our understanding of the social determinants of mental health: A data linkage study of mental health records and the 2011 UK census" Under the 'objectives' heading of the abstract, we clearly state that this paper described the linking of clinical records to the 2011 census due to the lack of socioeconomic information in clinical records (emphasis in bold): "To address the lack of individual-level socioeconomic information in electronic health care records, we linked the 2011 census of England and Wales to patient records from a large mental healthcare provider. This paper describes the linkage process and methods for mitigating bias due to non-matching." We are planning further analyses which will work with this linked data resource to shed light on broader social determinants, as the reviewer highlights, but the scope of this paper was to highlight the potential of data linkage, with methods to mitigate against bias, to improve our understanding of mental health inequalities.
3	It is suggested that males were less likely to be linked than women. Usually, the opposite is the case due to change of name following marriage etc. Was there any additional analysis undertaken to double-check this finding?	There is no way for us to determine if a person has changed their last name. It's true that this is a potential cause for records not matching. However, matchkey 4 was designed to address discrepancies in surnames, and it accounted for 4.1% of all matches (see Table 1). We also discuss this issue more generally and how the use of multiple matchkeys is intended to address this problem: "Because a single matchkey might be unable to resolve inconsistencies between data sources, multiple matchkeys were employed. Table 1 summarises each matchkey, the degree to which they uniquely identified records in each dataset, the proportion of CRIS to census matches, and the specific discrepancy they intended to address. For instance, matchkey 2 did not include postcode, thereby allowing records to match on name and date of birth, even if the individual's residence had changed." Page 6.
4	Perhaps the addition of GP records where more up-to-date information on address is available could improve linkage success rates. Would that be available through the electronic healthcare records?	We don't have access to GP records. It's worth noting however that using the most recent address to match records from CRIS to census may have not improved linkage success, since individuals will have been more likely to have a different address as more time elapses from the date of the census. Indeed, we observed the highest match rate for individuals whose first recorded contact with mental health services occurred in 2011 (see Figure 1).

5	There was no major difference when accounting for face-to-face contacts. This is perhaps because the severity of condition might vary significantly. Perhaps sensitivity analyses exploring the reason/type of visit could provide further details.	Table 2 suggests that with more face-to-face contacts CRIS cases were slightly more likely to match to census (Prevalence ratio of 1.10 (95% CI: 1.09-1.11) for 11+ contacts compared to reference group of no contacts). We would anticipate this to be the case as people with more face-to-face contacts have more opportunity to record key information that can improve the chances of successful data linkage. In table 3 we find that the weighted estimates for all-cause mortality in people with more frequent contacts is similar to the unweighted estimates. Unfortunately, we do not have accurate information in CRIS on the reason for a patients contact to undertake the analysis the reviewer suggests.
6	It would also be interesting to control for death by suicide, if the cause of death is available as differences will be expected in comparison to all-cause mortality.	For the current analysis we could not use cause-specific mortality and could only use all-cause mortality for analyses. We have added this as a limitation on page 12: "Finally, we could not examine cause-specific mortality, but will explore this in future analyses with linked data from the ONS mortality registration."
7	Please clarify what "service contact" might involve. Currently it is not clear to the reader.	We have clarified this on page 7: "We determined frequency of contact with services by counting the number of times they had been referred."
8	In the case of multiple diagnoses, how was the primary diagnosis chosen or were all diagnoses considered together?	We have clarified in the methods section that we used the diagnostic information that was present in the patient's "primary diagnosis" field. "When patients had multiple diagnoses, we used the information in the 'primary diagnosis' field." Page. 7

VERSION 2 – REVIEW

REVIEWER	Foteini Tseliou Cardiff University
REVIEW RETURNED	21-Sep-2023

GENERAL COMMENTS	The authors have provided additional information that can help address both reviewer comments. These responses/clarifications make it easy for the reader to understand the choice of data and variables made by the authors, but I believe that adding some of them in the main can strengthen the paper. I have two minor suggestions: 1. In the predictors of linkage subsection, it is noted that linkage rates vary by referral year. Could this perhaps be linked to changes in the processes followed (e.g. recording or transferring cases across healthcare settings)? I am wondering if this the case for all catchment areas within SLaM, as you added a mention that
--

	response rates from 88-94%. Perhaps add a quick note in the discussion. 2. Add 1-2 sentences on the caveats of the Census in investigating the association of interest e.g. based on self-report and subject to potential under-reporting, findings might apply to the respondents rather than the whole population, Census usually completed by one person in the household.
--	---

VERSION 2 – AUTHOR RESPONSE

	Comment	Response
	Reviewer 2	
1	In the predictors of linkage subsection, it is noted that linkage rates vary by referral year. Could this perhaps be linked to changes in the processes followed (e.g. recording or transferring cases across healthcare settings)? I am wondering if this the case for all catchment areas within SLaM, as you added a mention that response rates from 88-94%. Perhaps add a quick note in the discussion.	The proportion of linked records peaked in 2011, the year of the census, and diminished in preceding and subsequent years. We believe that this primarily was due to discrepancies in the recorded postcode in mental health records and in census due to migration because a), this piece of information was part of matchkeys that accounted for the majority of matches, and b), discrepancies in other key pieces of information used in these matchkeys, like sex, name, or date of birth, are less likely to change over time in the same manner. We make this point on page 11 in the discussion: “Moreover, because most matchkeys required postcode information to match and because the match rate peaked among individuals who were referred the year the census was taken, it is possible that the deterministic matching methodology that we employed also missed some individuals who had a different address at the time they interacted with SLaM services and responded to the census. This is supported by higher observed levels of matching (60%) for those with an address recorded in the mental health records at the time of census, in 2011, and is consistent with the interpretation that a high proportion of the sample in this study were potentially more mobile.” For this reason, we believe that it is unlikely that this general pattern of results is accounted for by changes in the recording or processing of basic demographic information like sex, age, or name. On the contrary, recording of some of these variables have remained stable or improved with time in healthcare settings, which is not consistent with a decrease in the match rate in the latter years of the study period. Because successful matches only occurred if all constituent parts of a matchkey (e.g., sex, dob, name, postcode) matched perfectly, it again points to discrepancies in the postcode being the primary driver of this trend.

2	Add 1-2 sentences on the caveats of the Census in investigating the association of interest e.g. based on self-report and subject to potential under-reporting, findings might apply to the respondents rather than the whole population, Census usually completed by one person in the household.	Census is completed by all members of the household, not by one person per household (see “Office for National Statistics. 2011 Census: Method and Quality Report. Response rates in the 2011 census. December 2012, published 16 July 2012”). We agree self-report could be a limitation leading to under-reporting for some elements, e.g., migration status or employment status, and so have added a sentence in the discussion on page 13 to reflect this. “Our study describes the process of linking Census to mental health electronic records. In the future, we plan to undertake assessments for the association of social and economic indicators from census with potential mental health outcomes. However, a limitation of Census is that it is self-report, and this may lead to under-reporting for some important indicators (e.g. migration status, employment status). This will need to be considered in future work.” Reference: Office for National Statistics. 2011 Census: Method and Quality Report. Response rates in the 2011 census. December 2012, published 16 July 2012 https://webarchive.nationalarchives.gov.uk/ukgwa/20160115211827/http://www.ons.gov.uk/ons/guide-method/census/2011/census-data/2011-census-user-guide/quality-and-methods/quality/quality-measures/response-and-imputation-rates/index.html